# Charged Amino Acids in the Transmembrane Helix Strongly Affect the Enzyme Activity of Aromatase

**DOI:** 10.3390/ijms25031440

**Published:** 2024-01-24

**Authors:** Juliane Günther, Gerhard Schuler, Elin Teppa, Rainer Fürbass

**Affiliations:** 1Research Institute for Farm Animal Biology (FBN), 18196 Dummerstorf, Germany; 2Veterinary Clinic for Reproductive Medicine and Neonatology, Faculty of Veterinary Medicine, Justus Liebig University, 35392 Giessen, Germany; gerhard.schuler@vetmed.uni-giessen.de; 3Université de Lille, CNRS, UMR 8576–UGDF-Unité de Glycobiologie Structurale et Fonctionnelle, 59000 Lille, France; elin.teppa@univ-lille.fr

**Keywords:** CYP19 isoforms, helix C, CPR-interacting surface, structural modeling, testosterone, androstenedione

## Abstract

Estrogens play critical roles in embryonic development, gonadal sex differentiation, behavior, and reproduction in vertebrates and in several human cancers. Estrogens are synthesized from testosterone and androstenedione by the endoplasmic reticulum membrane-bound P450 aromatase/cytochrome P450 oxidoreductase complex (CYP19/CPR). Here, we report the characterization of novel mammalian CYP19 isoforms encoded by CYP19 gene copies. These CYP19 isoforms are all defined by a combination of mutations in the N-terminal transmembrane helix (E42K, D43N) and in helix C of the catalytic domain (P146T, F147Y). The mutant CYP19 isoforms show increased androgen conversion due to the KN transmembrane helix. In addition, the TY substitutions in helix C result in a substrate preference for androstenedione. Our structural models suggest that CYP19 mutants may interact differently with the membrane (affecting substrate uptake) and with CPR (affecting electron transfer), providing structural clues for the catalytic differences.

## 1. Introduction

In virtually all vertebrates, from fish to mammals, estrogens are critical regulators of embryonic development, gonadal sex differentiation, behavior, and reproduction [1]. Estrogen-producing tissues are mainly the gonads, but also include the brain, adipose tissue, and, in humans and ungulates, the placenta [2,3].

The final step in estrogen biosynthesis, the conversion of testosterone (TST) or androstenedione (A4) to estradiol-17β (E2) or estrone (E1), is catalyzed by cytochrome P450 aromatase (CYP19). Human CYP19, a 503-amino acid protein, consists of a globular domain responsible for catalytic activity and an N-terminal transmembrane helix (TMH) that anchors it to the cytoplasmic surface of the endoplasmic reticulum (ER) membrane. Human aromatase has been particularly well studied in terms of its catalytic properties [4,5], molecular structure [6,7,8], and post-translational modifications such as phosphorylation [9] due to its role in estrogen-dependent cancers. The aromatase reaction consumes electrons provided by cytochrome P450 oxidoreductase (CPR). CPR also provides electrons to other microsomal CYP450 proteins in a highly competitive scenario [10,11]. A comparison of the aromatase sequences within the Gnathostomata (jawed vertebrates) also shows that the plasticity of the interaction with CPR is also highly relevant within the evolution of aromatase [12]. Key amino acids in the active site and substrate binding region are highly conserved evolutionarily. The interaction site of aromatase with CPR, on the other hand, allows for more variance and represents a region that may have been optimized during evolution. The emergence of phosphorylation sites in the CPR interaction region, which allow tissue-specific regulation, also seems to play a relevant role.

In humans, CYP19 is the product of a single *CYP19A1* gene [2], and it has been assumed that this is the case in all mammals. *CYP19A1* transcription in estrogen-producing tissues uses multiple tissue-specific promoters [2,13]. In contrast, certain fish (Teleostei) have two aromatase genes that encode a gonadal and a brain-specific isoform, respectively [14,15]. In cattle, a second *CYP19* gene was discovered for the first time in a mammal, but it turned out to be a pseudogene [16]. In pigs, three *CYP19* genes are known to encode ovarian, placental, and embryonic isoforms, respectively [17]. Unlike other aromatases, the ovarian isoform contains only 501 amino acids due to the absence of two amino acids in the N-terminal TMH [17,18]. Placental and ovarian isoforms exhibit catalytic differences [19,20,21], but the molecular or structural causes of these differences are not known.

Here, we report the catalytic and in silico structural characterization of previously unknown CYP19 isoforms identified by chance during a systematic database search for *CYP19* genes in terrestrial vertebrates.

## 2. Results

### 2.1. Artiodactyl Species Have More Than One CYP19 Gene

As part of another study, we performed a database search for *CYP19* genes in terrestrial vertebrates (amphibians, reptiles, and mammals) and found up to three *CYP19* genes only in the mammalian order Artiodactyla (even-toed ungulates). In this paper, we refer to the paralogous genes as *CYP19A1*, *CYP19A2*, and *CYP19A3*, and their products as CYP19A1, CYP19A2, and CYP19A3 isoforms. The *CYP19* coding sequences are included in Appendix A. Artiodactyls are classified into the suborders Suina (peccaries, pigs), Tylopoda (llamas, camels), Cetancodonta (hippos, whales), and Ruminantia (e.g., bovids, deer, mouse deer, giraffes), and all have *CYP19* paralogs. We therefore asked whether gene duplication occurred in a common ancestor or only after the diversification of the suborders. To this end, we analyzed the accumulation rates of synonymous mutations (dS) among *CYP19* gene pairs and found clear differences among the suborders, ranging from 0.14 in Ruminantia (mouse deer, *Tragulus kanchil*) to 0.04 in Tylopoda (*Camelus ferus*) (Table 1).

### 2.2. There Are Two Distinct Types of CYP19 Isoforms

Using 39 aligned CYP19 sequences from 24 terrestrial vertebrate species (Figure 1), we identified two structurally distinct types of CYP19 isoforms characterized by mutations at positions 42 and 43 in the N-terminal TMH, which always occur together with mutations at positions 146 and 147 in helix C of the catalytic domain. In most CYP19 sequences, these are mutations E42K, D43N and P146T, F147Y, whereas human CYP19A1 features E42, G43 and P146, F147 (all positions refer to human CYP19A1 sequence P11511, UniProtKB).

### 2.3. Isoform Type-2-Specific Mutations Affect the Catalytic Properties of Recombinant Human CYP19A1

To find out whether the N-terminal TMH and helix C mutations affect catalytic properties, we incubated recombinant CYP19A1 wild-type protein, CYP19A1 mutants E42K, D43N (CYP19A1_KN) and P146T, F147Y (CYP19A1_TY), and a CYP19A1 mutant with all four mutated residues (CYP19A1_TY+KN) with ^3^H-labeled androgens under standardized reaction conditions. At the end of the incubation period and after HPLC separation, we measured the ^3^H activity in the fractions corresponding to the remaining substrates and the estrogens produced. The measurements (expressed as a percentage of the ^3^H activities of the added substrates) are summarized in Figure 2.

In a first set of experiments, ^3^H-labeled TST was converted to E2. For CYP19A1, (rounded) 70% and 20% of the activity was associated with TST and E2, respectively. However, for CYP19A1_KN, 30% and 60% of the activity was allocated to TST and E2, respectively. After the reaction with CYP19A1_TY, 70% of the activity was associated with TST and 20% with E2. After the reaction with the CYP19A1_KN+TY mutant, we found 40% of the activity in TST and 40% in E2.

In a second set of experiments, ^3^H-labeled A4 was converted to E1. For CYP19A1, 60% of the activity was associated with A4 and 30% with E1. The CYP19A1_KN-catalyzed reactions resulted in 30% and 60% of the activity being allocated to A4 and E1, respectively. CYP19A1_TY resulted in 40% A4 activity compared to 40% E1 activity. With the CYP19A1_KN+TY mutant, we found 10% of the activity associated with A4 but 70% with E1. We also measured the ^3^H activity in unidentified molecules named UP1 and UP2 in association with substrate ^3^H-TST, as well as UP3 and UP4 in association with substrate ^3^H-A4. The proportion of ^3^H activity in the HPLC fractions corresponding to the UPs was below 2% in the baseline samples and increased up to 10% in the different aromatase reactions. The UPs showed extremely short retention times, so they probably did not interact with the HPLC column material at all. As the UPs were unlikely to be steroids, we did not investigate them further.

### 2.4. Structural Models of CYP19 Isoform Types

To investigate possible structural reasons for the observed catalytic properties of CYP19 isoform type 1 (E42G43 and P146F147) and CYP19 isoform type 2 (K42N43 and T146Y147), we built an in silico structural model of human CYP19A1 in complex with CPR (in the open conformation) in a membrane bilayer. According to our model, residues 42 and 43 are located in the N-terminal TMH at the interface region between the ER membrane and cytosol (Figure 3).

Residues 146 and 147 are part of helix C of the catalytic domain. In our structural model of the CYP19–CPR complex, both residues are located at the interaction interface with the FMN domain of CPR (Figure 3). The contact surfaces of both proteins are known to be electrostatically complementary [24], with the positive charge on CYP19 (Figure 4a) and the negative charge on CPR predominating. A possible phosphorylation of T146 and Y147 of isoform 2 could lead to a drastic change in the charge distribution of the CYP19 surface. While the side chain of P146 or T146 faces the interaction surface of the FMN domain of CPR, the benzene ring of F147 or Y147 faces the loop between helix I and helix H of aromatase. The phenolic hydroxyl group of Y147 would be able to form a hydrogen bond O-H∙∙∙O with the carboxylate side chain of D290, thereby strengthening the interaction between helix C and helix I (Figure 4b middle). Furthermore, the contact distance to R288 is significantly shortened by the tyrosine hydroxyl group of the TY mutant. The phosphorylation of Y147 would further reduce this distance and allow for a salt bridge that would strengthen the intramolecular connection between helix C and helices I and H.

## 3. Discussion

Artiodactyla are unique among terrestrial vertebrates in possessing two to three *CYP19* genes. They appeared about 65–70 million years ago (Mya). The suborders formed > 65 Mya (Suina) and 60–65 Mya (Tylopoda), and about 60 Mya, the most recent common ancestor (MRCA) of Cetancodonta and Ruminantia lived [25]. When did the *CYP19* gene duplicate, already in the MRCA of artiodactyls or only after the formation of the suborders? We addressed this question by examining the accumulation rates of synonymous differences (dS) between *CYP19* gene pairs. Because synonymous mutations do not alter the protein and are not subject to selection, dS values provide a measure of the time elapsed since gene duplication [26]. If *CYP19* had already duplicated in the MRCA of artiodactyls, then the resulting *CYP19* pairs should have accumulated roughly equal numbers of synonymous differences by today (i.e., dS values should be approximately the same). This is clearly not the case (Table 1). Rather, our data suggest that *CYP19* duplications occurred only after the suborders separated, i.e., at different time points (first in Ruminantia, last in Tylopoda).

Wild-type CYP19A1 is expressed in the gonads of all terrestrial vertebrates studied to date and, in mammals, in additional tissues such as adipose tissue, the brain, and the placenta. Thus, CYP19A1 must be capable of coping with different conditions (e.g., the type and availability of substrates, competition with other microsomal CYP450 enzymes for CPR) in estrogen-producing tissues. In this respect, CYP19A1 needs to be an all-rounder, whereas CYP19A2-type aromatases are specialized enzymes, as suggested by their preference for A4 as a substrate and their increased enzyme activity. Interestingly, CYP19_KN+TY mutants never replaced the wild-type enzyme, but underlying mutations only established in additional *CYP19* gene copies, implying that the wild-type CYP19A1 isoform is indispensable. However, the tissues expressing CYP19_KN+TY have not yet been identified, except for early porcine embryos and porcine placenta. Although we have carefully checked the CYP19 coding sequences, we cannot completely exclude the possibility that pseudogenes are present among the *CYP19A2* and *CYP19A3* paralogs studied in this work. However, the mutations characteristic of type 2 aromatases were selected during evolution (i.e., before pseudogenization) in all suborders of Artiodactyla. Therefore, our results regarding the catalytic properties of type 2 aromatases are not affected by the possible presence of pseudogenes.

Remarkably, only the CYP19A1 wild-type and the CYP19_KN+TY isoform (CYP19 type 2) occur naturally. The fact that type 2 aromatases have evolved several times further underscores the functional significance of the combined KN plus TY mutations. However, with the studies of recombinant CYP19A1_KN and CYP19A1_TY mutants, we were able to attribute the increase in enzyme activity of type 2 aromatases to the altered N-terminal TMH and the substrate preference of A4 to helix C of the catalytic domain.

The CYP19 E42K and G43N mutations affect the N-terminal TMH, which has been little studied so far. The bacterially expressed CYP19 enzymes used in biochemical experiments had truncated N-termini but were still catalytically active [4]. Subsequently, the N-terminal TMH was considered to be merely a membrane anchor with no relevance to the catalytic properties of CYP19. In addition, the available protein crystal structures of CYP19A1 in the Protein Data Bank are incomplete. At present, the computed AlphaFold structural model of the wild-type CYP19A1 (AF_AFP11511F1) is the only one that includes the N-terminal TMH. This next-generation protein structure prediction tool is particularly useful for proteins such as aromatase, for which no complete experimentally determined protein structure is currently available. We have modified the AlphaFold model to represent the K42N43 mutant. In our structural model of membrane-bound CYP19, amino acids E42 and G43 or K42 and N43 are located in the interface region between the ER membrane and cytosol. In the CYP19A1 isoform, the negatively charged γ carbonyl group of the E42 side chain is repelled by the negatively charged head groups of the membrane phospholipids. In contrast, in the CYP19A2 isoform, the positively charged ε amino group of K42 is attracted to the membrane. In this interaction, the long aliphatic side chain can dive deep into the hydrophobic core zone of the lipid bilayer, while the positively charged amino group is still attached to the phospholipid head groups. This mode of interaction at the membrane–water interface is also known as snorkeling [27,28]. Further, the CYP19 TMH contains an aromatic amino acid, tryptophan (W), at position 39. Many membrane proteins feature aromatic amino acids (an “aromatic belt”) in the portion of their TMH located in the interface region of the acyl–carbonyl groups of the lipid bilayer. Tryptophan in particular plays an important role in anchoring, stabilizing, and orienting membrane proteins in the lipid bilayer because of the unique physico-chemical properties of its side chain [27,29,30]. We hypothesize that the orientation of CYP19 in the ER membrane is largely determined by W39 on the one hand and E42 or K42 on the other. X-ray structural analyses by Ghosh et al. (2009) [6] have shown that the substrate access channel connecting the active site to the surface of the catalytic domain opens toward the membrane surface through which the substrate is delivered. It is therefore reasonable to assume that the K42 mutation in the N-terminal TMH results in an orientation of CYP19 in the ER membrane that facilitates substrate uptake. Coarse-grained molecular dynamics simulations by Mustafa et al. (2019) [31] suggested that wild-type CYP19A1 and TMH-truncated CYP19A1 may have only minor differences in the interaction of the globular domain with the membrane. However, it should be noted that these studies were performed with an artificial, neutral POPC phosphatidylcholine bilayer.

In general, the N-terminal TMHs of CYP450 enzymes show a distribution of nonpolar, aliphatic residues (in the hydrophobic core region of the lipid bilayer), aromatic residues (in the acyl–carbonyl region), and polar charged and hydrophilic residues (in the membrane–water interface region) that is typical for membrane proteins [27,30]. However, CYP19A1 is the only one of 41 human microsomal CYP450 enzymes examined that has a negatively charged side chain (E42) in the membrane–water interface region (Appendix A). In all terrestrial vertebrate species that have solely the CYP19A1 isoform, at least one acidic amino acid is located at positions 42 and 43. This is apparently an essential feature of the wild-type CYP19A1 isoform, but why this is so remains to be elucidated.

According to our structural model of the CYP19–CPR complex, mutations P146T and F147Y affect the interaction interface with the FMN domain of CPR. Both proteins are electrostatically complementary at their contact surfaces, with a predominantly positive charge on CYP19 and a negative charge on CPR. In CYP19A2, the T146 and Y147 mutations have introduced possible phosphorylation sites in helix C. Phosphorylation sites typically serve as switches for the rapid regulation of enzyme activity. The addition of negatively charged phosphate would alter the electrostatic potential of the contact surface and significantly affect the interaction with CPR and electron transfer from FMN to heme. Given the strong competition with many microsomal CYP450 enzymes and the few available CPR molecules, this is likely to have an impact on aromatase activity. The phosphorylation of human CYP19A1 at various sites and the effects on enzyme activity have been previously reported, but there is no information regarding the effects of phosphorylating helix C [9,32,33]. However, other microsomal CYP450 enzymes have phosphorylation sites in their helix C, and the phosphorylation of these sites regulates the catalytic activity of the enzymes [11]. We therefore hypothesize that this is also true for T146 and Y147 of the CYP19A2 isoform.

Helix C is part of the contact surface of CYP19 with its redox partner CPR. In addition, the amino acids W141 and R145 of helix C interact with the propionate moieties of heme, the acceptor for electron transport from the FMN domain of CPR. Studies of the bacterial CYP450 enzyme P450BM3 have shown that binding between the FMN and heme domains and subsequent electron transfer is a highly dynamic system in which helix C plays a critical role. The helix C of CYP450 proteins is a very flexible structure. However, upon substrate binding, the interaction between helix I (which is involved in substrate binding) and helix C is strengthened, and the rotation of helix C is blocked in a position that facilitates electron transfer. This allows for a sufficiently tight contact between helix C and the FMN domain, which is necessary for efficient electron transfer to heme [34]. In CYP19A1, P146 should cause a kink in the helix C structure because it cannot form stabilizing hydrogen bonds to the carbonyl groups of the K142 and K143 side chains. Interestingly, no other CYP450 enzyme carries a conserved proline in the center of its helix C (Appendix A), suggesting a functional role for P146. In contrast, the P146T mutation in CYP19A2 restored helix-stabilizing hydrogen bonds. In particular, binding stabilization at K142 may be important because other CYP450 proteins carry K or R residues at the corresponding position, which are known to form salt bridges to the FMN domain via E residues located at the end of the α1 helix [23,34]. We hypothesize that such interactions may also affect the binding between CYP19 and the FMN domain and, thus, electron transfer.

Another aspect of T146 phosphorylation is the conformational change it induces in the side chain; due to non-covalent interactions of the phosphate, the p-threonine (but not the p-serine) side chain prefers a cyclic conformation similar to the proline side chain [35]. In the absence of helix-stabilizing hydrogen bonds, pT146 could also induce a kink in helix C that could be regulated by kinase/phosphatase activities (in contrast to the P146 kink of the wild-type isoform). The most common phosphorylation sites in proteins are serine and threonine residues [35]. Although proline codons (CCN) should mutate to serine (UCN) or threonine (ACN) codons with equal probability, we did not find any S146 mutants. The T146 mutation, on the other hand, has been selected several times in Artiodactyla. Possibly, the introduction of a phosphorylation site at position 146 was important in the evolution of the CYP19A2 isoform, but at the same time, the maintenance of a proline-like structure in helix C.

CYP19 mutants with TY in helix C have shown higher activity than the wild-type CYP19A1 isoform with PF only with A4 as the substrate. The reason for the substrate preference of the TY mutant is not easily explained because helix C does not directly interact with the substrate. As mentioned above, the interaction between helix I and helix C after substrate binding is important for the stabilization of helix C and thus for an optimized interaction with the FMN domain. In the case of CYP19 isoform 2, the hydroxyl group Y147 could enhance the interaction between helices I and C, which could be further enhanced via the phosphorylation of Y147. Furthermore, substrate binding has also been shown to affect the conformational dynamics of the CYP19A1 protein. Paco et al. (2020) [36] observed a decreased mobility of helix C after A4 binding in hydrogen–deuterium exchange experiments. In another study by Hong et al. (2007) [37], helix C of CYP19A1 was protected from trypsin fragmentation after A4 binding. Unfortunately, similar studies with TST as a substrate are not available. However, studies on the structural dynamics of CYP3A4 suggest that the type of substrate indeed has an effect on the rigidification of the C-helix [38]. Looking at their results in detail, the hydrogen–deuterium exchange profiles show that the C-helix of CYP3A4 is not protected, i.e., not rigidified, after the binding of TST (Supplementary Data of Ducharme et al. 2021) [38]. In contrast, the binding of 7-benzyloxy-4-trifluoromethylcoumarin (BFC) and progesterone leads to a rigidification of the C-helix. Interestingly, both progesterone and BFC have keto groups, as does A4, whereas TST has a polar hydroxyl group in addition to the keto group.

However, these results should be interpreted with caution and several limitations should be noted. First, we cannot predict whether the newly identified CYP19 isoforms are actually expressed in Artiodactyla. Although we have found complete, intact open reading frames in most species, it is not known whether they are expressed as functional proteins and, if so, in which tissues. Nevertheless, type 2 aromatases with these specific TMH and helix C mutations appear to have evolved multiple times independently in each artiodactyl suborder. Whether or not they were pseudogenized later in evolution, this is evidence for the importance of these mutations for aromatase function. This is supported by the fact that the mutations significantly affect enzyme activity even when introduced into recombinant human CYP19A1.

We did not measure enzyme activity through the commonly used tritium–water assay (via the release of ^3^H water during the aromatase reaction), but separated the steroids chromatographically after the enzymatic reaction and determined the ^3^H activity of the individual peaks. This approach is both more sensitive and more specific, since the decrease in the substrate can be detected simultaneously with the increase in the specific metabolites. Our measurements provide initial but clear indications of the different enzyme properties of the two aromatase isoform types. However, for a more precise biochemical characterization, further analyses, such as the determination of V_max_ and K_m_, should be performed in future studies. It would also be interesting to investigate whether the TY residues in helix C of the type 2 enzymes are actually phosphorylated. This may also depend on the tissue in which the aromatases are expressed. Mass spectrometric methods could be used for this purpose.

## 4. Materials and Methods

### 4.1. Mutagenesis of Human CYP19A1 cDNA

The expression vector SC127848, containing human *CYP19A1*, transcript variant 1 (NM_000103), was obtained from OriGene Technologies (Herford, Germany). Site-directed mutagenesis was performed with the Q5 Site-Directed Mutagenesis Kit (New England Biolabs, Frankfurt am Main, Germany) according to manufacturer’s instructions. The following mutagenic primers (Merck, Darmstadt, Germany) were used to generate the KN and TY mutant, as well as the KN/TY double mutant: 5′-GTGGAATTATaagaacACATCCTCAATACCAG-3′ (sense primer for E42K, G43N mutation), 5′-ACCAAGAGAAAAAGGCCAG-3′ (antisense primer for E42K, G43N mutation), 5′-AACAACTCGAacctacTTTATGAAAG-3′ (sense primer for P146T, F147Y mutation), and 5′-TTCCAGAGCTCTGGATTG-3′ (antisense primer for P146T, F147Y mutation). Sanger sequencing was used to verify the substitution of the respective nucleotides (Microsynth, Göttingen, Germany). Plasmids from positive clones were prepared endotoxin-free for subsequent transfection using the EndoFree Plasmid Maxi Kit (Qiagen, Hilden, Germany).

### 4.2. Transfection and Transient Expression of Recombinant CYP19 in HEK293 Cells

HEK293 cells were cultivated in high-glucose DMEM (4.5 g/L), with stable glutamine supplemented with 10% FCS and a Penicillin/Streptomycin antibiotic mixture (Capricorn Scientific, Ebsdorfergrund, Germany). Wild-type CYP19A1, as well as KN, TY, and KN+TY mutant expression plasmids, were transfected using the Xfect Transfection Reagent (Takara Bio Europe SAS, Saint-Germain-en-Laye, France) following the manufacturer’s instructions. Cells were harvested 48 h after transfection and washed three times with PBS, and the cell pellet was subsequently stored at −70 °C.

### 4.3. Measurement of CYP19 Activity

The reaction mixture, totaling 100 μL, contained 2 μL of the ^3^H-labeled substrate, 5 μL of the respective recombinant enzyme, 25 μL of 1 mmol/L NADPH (Roche Diagnostics, Mannheim, Germany), and 68 μL of Dulbecco’s phosphate-buffered saline, pH 7.4 (Merck KGaA, Darmstadt, Germany), as incubation medium. The substrate concentrations in the reaction mixtures were 2.4 nmol/L for androstenedione ([1,2,6,7-3H(N)]-androstenedione, American Radiolabeled Chemicals, St. Louis, MO, USA) or 1.9 nmol/l for testosterone ([1,2,6,7-3H(N)]-testosterone, PerkinElmer Health Sciences, Groningen, The Netherlands). For negative controls, the recombinant enzyme was replaced by water (“baseline”) or a preparation of non-transfected cells (“HEK_control”). The samples were incubated for 30 min at 37 °C. At the end of the incubation period, the reaction was stopped by adding 2 mL of toluene. The samples were then extracted twice with 2 mL of toluene and the dried extracts were redissolved in 50 μL of HPLC mobile phase (H_2_O/methanol/tetrahydrofuran 60/32/8 (*v*/*v*/*v*)). Preliminary experiments confirmed the linearity of the conversions over the incubation period. To measure substrate conversion, 20 μL aliquots of the sample were analyzed via HPLC on a reversed-phase column (150 × 4 mm Eurospher II 100-5 C18, Knauer GmbH, Berlin, Germany) under isocratic conditions at a flow rate of 1.0 mL/min. The eluate was collected in 1.0 mL fractions, in which the ^3^H activity was measured in a β-counter (Tri-Carb 2810 TR PerkinElmer, Rodgau, Germany) after the addition of a 3 mL liquid scintillation fluid (Rotiscint eco plus, Carl Roth GmbH + Co. KG, Karlsruhe, Germany). The substrate and metabolite were identified based on a comparison of their retention times with authentic tritiated standards. The determination of estrogen formation was based on the distribution of the ^3^H activity across peaks for each substrate and its product. The final results were calculated from three independent experiments.

### 4.4. Structural Modeling of Aromatase Isoforms

All visualizations of the aromatase protein model were generated using UCSF Chimera [39] and ChimeraX [40]. The structural model of human aromatase including TMH (UniProt P11511) was obtained from the AlphaFold database [41]. It was generated using the AlphaFold Monomer v2.0 pipeline [42]. We incorporated heme and androstenedione in the active site of this aromatase model. Their positions correspond to that of the crystal structure of the globular aromatase domain 3EQM. The interaction between human aromatase and the FMN domain in the open conformation of human CPR was modeled using the multiresolution simulations of the CYPA1A–CPR complex (ModelArchive doi:10.5452/ma-3oc22). Despite the low sequence identity between CYP19A1 and CYPA1A (20% sequence identity), they show a good structural superposition, with an average C-alpha RMSD = 2.135 Å calculated over 381 aligned residues. No steric clashes were observed in the resulting CYP19A1-CPR model. The KN and TY mutations in the model structure of human aromatase, as well as the incorporation of phosphate groups at TY, were performed using FoldX5 [43].

### 4.5. Capillary-Based Western Analysis, Primary Antibodies

For normalization purposes, we examined the aromatase levels in lysates of transfected KEK293 cells via a Western analysis using the Protein Simple Wes System (Bio-Techne, Wiesbaden, Germany), as previously described [44]. The anti-CYP19A1 mouse monoclonal antibody raised against polypeptide AA376-390 of human CYP19A1 (clone H4, SM2222P, Acris Antibodies, Herford, Germany) and the anti-ACTB mouse monoclonal antibody against chicken beta-actin (sc-47778, Santa Cruz Biotechnology, Inc., Dallas, TX, USA) served as the primary antibodies.

### 4.6. Estimation of the Accumulation Rates of Synonymous Mutations

We determined the accumulation rate of synonymous mutations (dS = S/ES) between pairs of paralogous *CYP19* genes to provide a measure of the time elapsed since *CYP19* gene duplication. S is the number of synonymous differences between two *CYP19* genes and ES is the number of expected (i.e., all possible) synonymous sites in a gene. We computed the number of synonymous mutations using the method of Nei and Gojobori (1986) [45]; to determine the ES of a *CYP19* gene, we determined the number of possible synonymous mutations for each individual codon and summed them across all codons. When comparing *CYP19* gene pairs, we averaged the individual ES values. To determine S, we counted the synonymous differences between *CYP19* gene pairs. The determination of synonymous mutations in a codon was straightforward as long as only a single nucleotide substitution was present. In cases where more than one mutation was required to convert one codon to another, we analyzed the possible pathways and averaged the number of synonymous mutations of all pathways.

A flow chart illustrating the methodology is provided in Appendix A.

## 5. Conclusions

In summary, we have identified the key regions in the protein structure characteristic of the two mammalian aromatase isoform types. It is noteworthy for further (evolutionary biological) research that isoform type 2 is exclusive to artiodactyls, in addition to the ubiquitous isoform type 1. Furthermore, by functionally characterizing these regions, we have gained new insights into the structure–function mechanism of aromatase that are likely to be relevant to other microsomal CYP450 enzymes. We show that the charge of TMH at the contact surface between the ER membrane and the cytoplasm is critical for the activity level of the enzyme. In addition, the structure of helix C, although not part of the catalytic pocket and thus not directly involved in substrate binding, significantly affects the substrate preference of the enzyme. Both regions are accessible on the surface of the protein and could represent potential targets for inhibitors and, thus, new therapeutic approaches, e.g., against estrogen-dependent cancers.

## Figures and Tables

**Figure 1 ijms-25-01440-f001:**
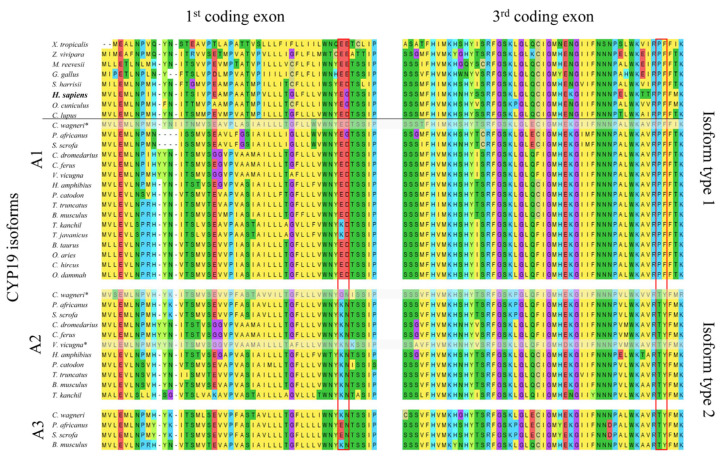
Comparison of the amino acid sequences of aromatases encoded by the first and third coding exons of *CYP19* genes from selected tetrapods. The horizontal line separates species with only one *CYP19* gene (**top**) from Artiodactyla species with two to three *CYP19* genes (**bottom**). All Artiodactyla express only two CYP19 isoform types, including those with three *CYP19* genes (Suina (pigs and peccaries); Mysticeti (baleen whales)). The amino acid residues defining the two isoforms are outlined in red. Amino acids are labeled according to the MegaX [22] color code. The following NCBI protein sequences were used: *Xenopus tropicalis* NP_001090630.1, *Zootoca vivipara* XP_034986371.1, *Mauremys reevesii* XP_039348262.1, *Gallus gallus* NP_001001761.3, *Sarcophilus harrisii* XP_031811056.1, *Homo sapiens* NP_000094.2, *Oryctolagus cuniculus* NP_001164392.1, and *Canis lupus familiaris* NP_001008715.1. All Artiodactyla sequences are included in Appendix A. * Not all of the nine coding exons were found in the available databases.

**Figure 2 ijms-25-01440-f002:**
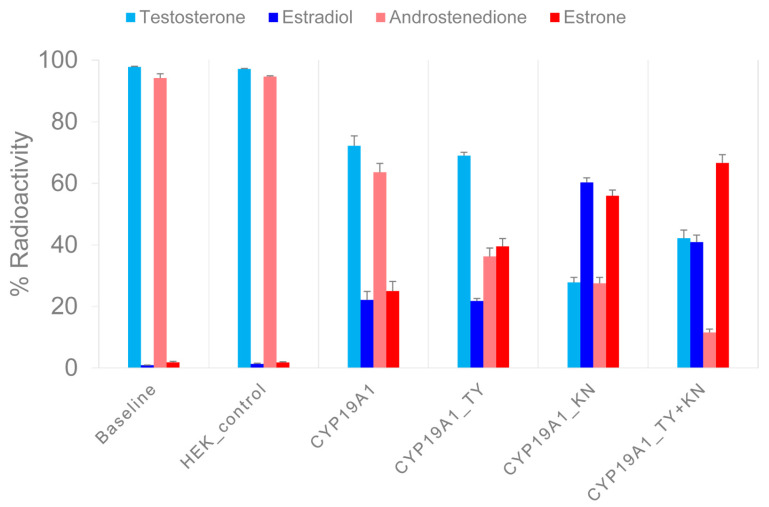
Measurement of enzymatic activity of recombinant CYP19A1 wild-type and mutant proteins. In two sets of experiments, ^3^H-testosterone and ^3^H-androstenedione were converted to ^3^H-estradiol and ^3^H-estrone, respectively, under standardized reaction conditions. After a reaction time of 30 min, the products were separated via HPLC and the ^3^H activity was measured in the collected fractions. The *X*-axis shows the mean values of the ^3^H activity in the androgen substrates and in the estrogen products of three experiments, each as a percentage of the input ^3^H activity. Error bars indicate standard errors of the mean.

**Figure 3 ijms-25-01440-f003:**
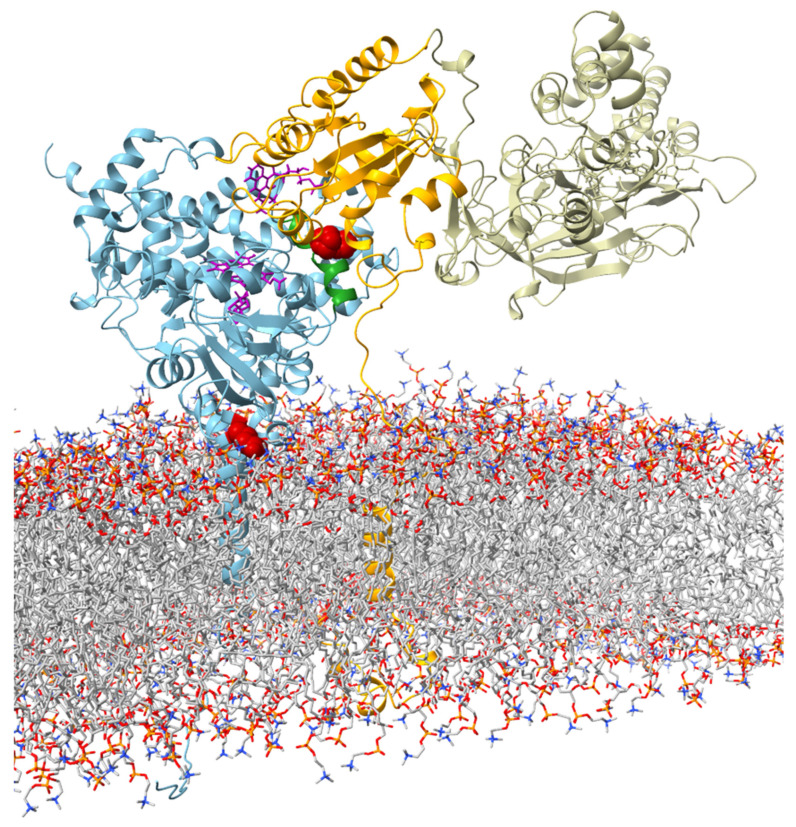
Visualization of the interaction between aromatase and CPR (open confirmation adapted from Mukherjee et al. (2021) [23]) at the ER membrane. The protein backbone of aromatase is shown in blue. The amino acids E42G43 in the membrane–water interface region and P146F147 in the C-helix are highlighted in red. The C-helix is colored in green. FMN and heme are shown in violet in stick representation. The subunits of CPR are color-coded (orange: TMH, linker region, and FMN; tan: FAD and NADP).

**Figure 4 ijms-25-01440-f004:**
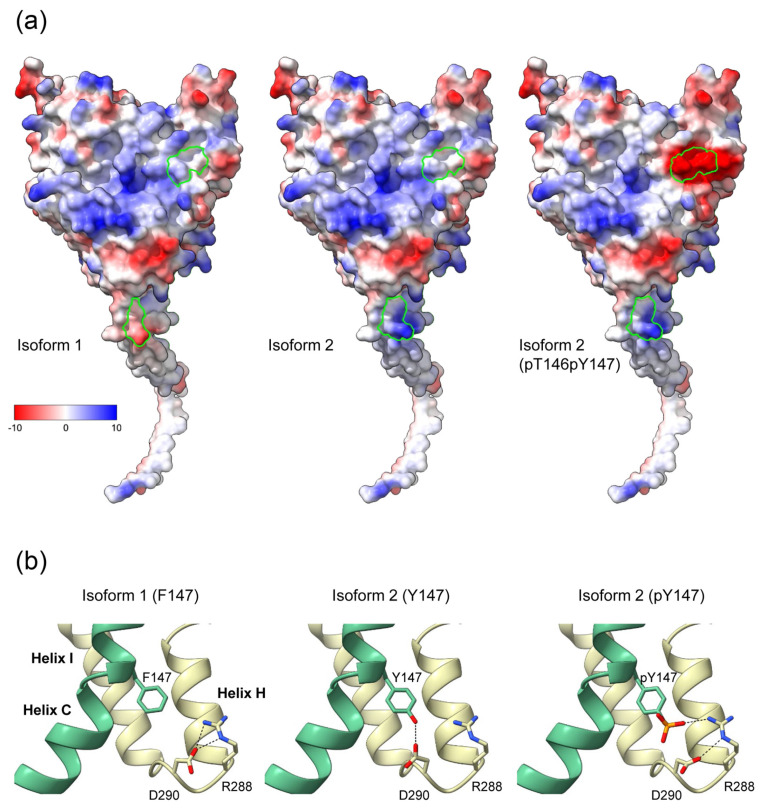
Effects of the K42N43 and T146Y147 mutations on the electrostatic surface potential and intramolecular interactions of aromatase. (**a**) Surfaces of CYP19 isoforms facing the FMN domain of CPR, colored according to electrostatic potential. The Coulombic charge distribution ranges from negative (red) to positive (blue) are indicated. Amino acids 42 and 43 at the end of the TMH and 146 and 147 of helix C of isoform 1, isoform 2, and isoform 2 with phosphorylated T146 and Y147 are circled in green. (**b**) Possible intramolecular interaction of F147 or Y147 with R288 and D290 in the loop connecting helix H and I. Helix C is colored green. Black dashed lines indicate possible hydrogen bond interactions in the different isoforms.

**Table 1 ijms-25-01440-t001:** Accumulation rates (dS) of synonymous mutations between paralogous *CYP19* gene pairs of artiodactyl species.

	Species			
	*Ssc*	*Cfe*	*Bmu*	*Tka*
S	38.00	13.00	19.50	49.50
ES	335.50	332.67	335.17	342.33
**dS**	**0.11**	**0.04**	**0.06**	**0.14**

S, synonymous differences; ES, expected synonymous sites; dS = S/ES; *Ssc*, *Sus scrofa* (Suina); *Cfe*, *Camelus ferus* (Tylopoda); *Bmu*, *Balaenoptera musculus* (Cetancodonta); *Tka*, *Tragulus kanchil* (Ruminantia).

## Data Availability

Data are contained within the article.

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
