# Peer review of "Charged Amino Acids in the Transmembrane Helix Strongly Affect the Enzyme Activity of Aromatase"

_ijms, 2024, doi:10.3390/ijms25031440_

Round 1

Reviewer 1 Report

Comments and Suggestions for Authors

In this study, Gunther et al. provide a detailed analysis of newly discovered mammalian CYP19 isoforms that are encoded by copies of the CYP19 gene. All of these CYP19 isoforms are characterized by a combination of mutations in the N-terminal transmembrane helix and in helix C of the catalytic domain. The mutant CYP19 isoforms exhibit enhanced androgen conversion as a result of the KN transmembrane helix. The structural models indicate that CYP19 mutations may have distinct interactions with the membrane, which might impact the absorption of substrates, as well as with CPR, which could alter the transfer of electrons. These structural findings provide valuable insights into the variations in catalytic activity. Throughout the content, the authors have successfully encapsulated the dynamic interplay between structure and function of enzymes, underscoring its potential to revolutionize the field. This article promises to offer valuable insights into the rapid developments in structure-function understanding, emphasizing the importance of both technological advancement and the growing popularity of structural bioinformatics.

Overall, the paper is well written and the discussion in particular is interesting. Although the authors fail to introduce some basic concepts in the introduction section, I feel they are important and an addition to the existing literature. I recommend the paper for publication after addressing some of my concerns.

My first concern, the authors miss some important citations. They failed to discuss the literature already published (https://doi.org/10.3390%2Fijms22020631; https://doi.org/10.1016/j.bpj.2018.12.014; and other).

In particular, I wonder how the authors miss to cite some classical papers about the enzyme kinetics.

It is difficult to grasp the overall direction of the paper. The authors should add a flowchart that schematically represents overall methodology adopted in the paper.

The limitation of this study should also be discussed.

The paper is lack of detailed summary of the advantages and disadvantages of the study presented.

Since the conclusions stated highly depend on programs such as Alphafold, the authors should praise those popular methods in the introduction section (instead of just citing them).

Author Response

Dear Reviewer,

Thank you very much for reviewing our manuscript. Your advice helped us to improve our manuscript significantly. We have highlighted our changes in red in the revised version.

In the following, we would like to respond in detail to your queries.

  • My first concern, the authors miss some important citations. They failed to discuss the literature already published (https://doi.org/10.3390%2Fijms22020631); https://doi.org/10.1016/j.bpj.2018.12.014; and other).

Thanks for drawing this to our attention. We now cite the articles by Di Nardo et al. 2021 and Mustafa et al. 2019 in our manuscript (new references 12 and 31). In addition, we have made the following changes to the text in order to discuss these articles (lines 40-47 and 242-246 of the revised manuscript).

Lines 40-47: “A comparison of the aromatase sequences within the gnathostomata (jawed vertebrates) also shows that the plasticity of the interaction with CPR is also highly relevant within the evolution of aromatase [12]. Key amino acids in the active site and substrate binding region are highly conserved evolutionarily. The interaction site of aromatase with the CPR, on the other hand, allows for more variance and represents a region that may have been optimized during evolution. The emergence of phosphorylation sites in the CPR interaction region, which allow tissue-specific regulation, also seems to play a relevant role.

Lines 242-246: “Coarse-grained molecular dynamics simulations by Mustafa et al. (2019) [31] suggested that wild-type CYP19A1 and TMH-truncated CYP19A1 may have only minor differences in the interaction of the globular domain with the membrane. However, it should be noted that these studies were performed with an artificial, neutral POPC phosphatidylcholine bilayer.

  • In particular, I wonder how the authors miss to cite some classical papers about the enzyme kinetics.

Since we did not investigate aromatase enzyme kinetics in our studies, we did not explicitly discuss the classic articles dealing with enzyme kinetics.

  • It is difficult to grasp the overall direction of the paper. The authors should add a flowchart that schematically represents overall methodology adopted in the paper.

Thank you for your suggestion. We have added a new Figure S3: "Flow chart illustrating methodology and study design" to the manuscript.

  • The limitation of this study should also be discussed.
  • The paper is lack of detailed summary of the advantages and disadvantages of the study presented.

Thank you for your advice. We regret that we did not discuss the limitations, advantages and disadvantages of our study in the first version of our manuscript. We have now addressed these issues in detail in the revised version (lines 323-344):

However, these results should be interpreted with caution and several limitations should be noted. First, we cannot predict whether the newly identified CYP19 isoforms are actually expressed in Artiodactyla. Although we found complete intact open reading frames in most species, it is not known whether they are expressed as functional proteins and, if so, in which tissues. Nevertheless, type 2 aromatases with these specific TMH and helix C mutations appear to have evolved multiple times independently in each artiodactyl suborder. Whether or not they were pseudogenized later in evolution, this is evidence for the importance of these mutations for aromatase function. This is supported by the fact that the mutations significantly affect enzyme activity even when introduced into recombinant human CYP19A1.

We did not measure enzyme activity by the commonly used tritium-water assay (via the release of 3H water during the aromatase reaction), but separated the steroids chromatographically after the enzymatic reaction and determined the 3H activity of the individual peaks. This approach is both more sensitive and more specific, since the decrease of the substrate can be detected simultaneously with the increase of the specific metabolites. Our measurements provide initial but clear indications of the different enzyme properties of the two aromatase isoform types. However, for a more precise biochemical characterization, further analyses, such as the determination of Vmax and Km, should be performed in future studies. It would also be interesting to investigate whether the TY residues in helix C of the type 2 enzymes are actually phosphorylated. This may also depend on the tissue in which the aromatases are expressed. Mass spectrometric methods could be used for this purpose.

  • Since the conclusions stated highly depend on programs such as Alphafold, the authors should praise those popular methods in the introduction section (instead of just citing them).

Thank you for your suggestion. We have now explicitly emphasized the importance of AlphaFold not only for our study in lines 219-221:

This next-generation protein structure prediction tool is particularly useful for proteins such as aromatase, for which no complete experimentally determined protein structure is currently available.”

Reviewer 2 Report

Comments and Suggestions for Authors

This manuscript reports a preliminary and partial characterization of putative isoforms of aromatase in Artiodactyl species.

The catalytic activity of the enzymes was measured in cells and a structural model was constructed and docked to CPR. The data shown in the paper are very few and there are some major concerns about them. The discussion section is very long and detailed and not fully supported by the data, especially by the predictions. Moreover, the relevant work in the literature about aromatase is only partially acknowledged.

I think this work should be improved in order to be published in IJMS and relevant functional experiments should be added according to the following issues:

Line 51. The “structural characterization” mentioned is not actually present in the work as there is not a solved structure but in silico predictions.

Line 52. Can the Authors exclude that the new isoforms identified are not pseudogenes? Is there any evidence of their expression?

Lines 74. From here, I do not understand how many isoforms are studied and how they are called in the rest of the manuscript. In Figure 2, it looks like there are 3 isoforms or the mutations are separated and then combined?

Lines 120-123. This paragraph is not clear and nothing is mentioned in the Methods section.

Figure 3. This model looks very strange as the CPR seems to be not attached to the membrane. The Authors should carefully check what is published in the literature and what is known about aromatase interaction with CPR (doi: 10.3390/ijms22020631, DOI: 10.1016/j.ijbiomac.2020.07.163, https://doi.org/10.1016/j.jsbmb.2009.11.010).

Line 239. Reference 9 is not the only work demonstrating phosphorylation in human aromatase (see doi DOI: 10.2174/1389557516666160321113041, https://doi.org/10.1016/j.jsbmb.2008.09.001 and  DOI: 10.1016/j.jsbmb.2016.09.022).

Lines 318-320. The final concentration should be mentioned, it does not make sense to indicate volumes. The substrate concentrations indicated for the activity measurements are 2.4 and 1.9 nM. Such concentrations are far away from the saturating ones and also from the Km (at least for the human enzyme). How was this concentration chosen? If the CYP19 isoforms have different Km, then isoforms will have different rates at different substrate concentration. The experiment shown in Figure 2 would make sense if the concentration is saturating for all the compared enzymes.

Lines 338-339. I do not understand here what the meaning of this sentence is.

Author Response

Dear Reviewer,

Thank you very much for reviewing our manuscript. We hope that we were able to address your suggestions sufficiently to improve the manuscript. We have highlighted our changes in red in the revised version.

In the following, we would like to respond to your questions and comments in more detail.

  • Line 51. The “structural characterization” mentioned is not actually present in the work as there is not a solved structure but in silico predictions.

Thank you for bringing this to our attention. We have now clarified in lines 58 and 140 that these are in silico models.

  • Line 52. Can the Authors exclude that the new isoforms identified are not pseudogenes? Is there any evidence of their expression?

No, we cannot exclude the possibility that some of the newly identified isoforms are pseudogenes. Also, it is not known where these isoforms are expressed in the different Artiodactyla species, except for the pig. We have now addressed this point in the revised version of the manuscript (lines 199-205 and 323-332).

Lines 199-205: “Although we have carefully checked the CYP19 coding sequences, we cannot completely exclude the possibility that pseudogenes are present among the CYP19A2 and CYP19A3 paralogs studied in this work. However, the mutations characteristic of type 2 aromatases were selected during evolution (i.e., before pseudogenization) in all suborders of Artiodactyla. Therefore, our results regarding the catalytic properties of type 2 aromatases are not affected by the possible presence of pseudogenes.

Lines 323-332: “First, we cannot predict whether the newly identified CYP19 isoforms are actually expressed in Artiodactyla. Although we found complete intact open reading frames in most species, it is not known whether they are expressed as functional proteins and, if so, in which tissues. Nevertheless, type 2 aromatases with these specific TMH and helix C mutations appear to have evolved multiple times independently in each artiodactyl suborder. Whether or not they were pseudogenized later in evolution, this is evidence for the importance of these mutations for aromatase function. This is supported by the fact that the mutations significantly affect enzyme activity even when introduced into recombinant human CYP19A1.

  • Lines 74. From here, I do not understand how many isoforms are studied and how they are called in the rest of the manuscript. In Figure 2, it looks like there are 3 isoforms or the mutations are separated and then combined?

We apologize for not being clear in our description of these points. To improve the understanding of our statements, we have now optimized Figure 1 and precisely labeled the isoforms and isoform types. In addition, we have reworded the title 2.3. to sharpen it with respect to the results presented. The new title is now “2.3. Isoform type-2 specific mutations affect the catalytic properties of recombinant human CYP19A1”. In addition, we have provided a more detailed description of the aromatase proteins investigated in this result section (lines 104-106).

“…we incubated recombinant CYP19 wild-type protein, CYP19A1 mutants E42K+D43N (CYP19A1_KN) and P146T+F147Y (CYP19A1_TY) and a CYP19A1 mutant with all four mutated residues (CYP19A1_TY+KN)…”

  • Lines 120-123. This paragraph is not clear and nothing is mentioned in the Methods section.

We regret that in the first version of our manuscript we did not go into enough detail about the unknown molecules, called UPs, that we observed. We did not determine enzyme activity by the tritium-water assay, which only detects the formation of tritium-labeled water, but separated and analyzed both the radiolabeled substrate and the radioactive products after incubation with the CYP19 proteins by HPLC. This allows us to specifically determine both the substrate degradation caused by aromatase activity and the formation of the products. The androstenedione, testosterone, estrone and estradiol peaks can be identified based on known retention times. In addition, other peaks were identified, which we called UPs. For the sake of completeness, we did not want to leave this unmentioned. However, we agree that the description was incomplete and could lead to misunderstandings. Therefore, we have added information about this in the revised version (lines 134-136):

The UPs showed extremely short retention times, so they probably did not interact with the HPLC column material at all. As the UPs were unlikely to be steroids, we did not investigate them further.

  • Figure 3. This model looks very strange as the CPR seems to be not attached to the membrane. The Authors should carefully check what is published in the literature and what is known about aromatase interaction with CPR (doi: 10.3390/ijms22020631, DOI: 10.1016/j.ijbiomac.2020.07.163, https://doi.org/10.1016/j.jsbmb.2009.11.010).

We regret that the attachment of CPR to the membrane was not clearly visible in our Figure 3. We have now colored the transmembrane helix, the FMN domain and the linker region connecting both domains in orange. We hope that this will increase the visibility of the TMH in the membrane. Regarding the interaction of CRP with aromatase: There are no crystal structures describing this adduct. Unfortunately, the current assumptions about how aromatase and the FMN domain of the CPR might interact are based only on calculations with different Docking programs. Depending on the Docking program used, different results are obtained (Ritacco et al. 2020, https://doi.org/10.1021/acs.jpclett.9b03798). In addition, phosphorylation of aromatase also seems to change the orientation of aromatase to FMN (Y361, Ritacco et al. 2019, reference [11]). This shows that it is still unknown which amino acids of aromatase and the FMN domain interact with each other. However, the region of the aromatase protein that interacts with the FMN domain is known. Helix C is part of this region. It should be noted that the interaction and thus the electron transport only takes place in the open conformation of CPR, in which the orientation of the soluble CRP domains to the membrane also changes. Xia et al. 2019 (https://doi.org/10.1021/acs.biochem.9b00130) show that, in particular, the flexible linker region between the TMH and FMN domains plays an important role in facilitating the movement of the soluble domain relative to the membrane. This allows for multiple orientations that allow for specific interactions of the FMN domain with its different partners. The model we used for this CPR conformation comes from Mukherjee et al. (2021) [23]. We have now explicitly mentioned this in the legend of Figure 3.

  • Line 239. Reference 9 is not the only work demonstrating phosphorylation in human aromatase (see doi DOI: 10.2174/1389557516666160321113041 https://doi.org/10.1016/j.jsbmb.2008.09.001 and  DOI: 10.1016/j.jsbmb.2016.09.022

Thank you for this advice. We have now cited the research articles by Miller et al. 2008 (new reference [32]), which describes the phosphorylation of S108, and Baravalle et al. 2016 (new reference [33]), which demonstrates the importance of R264 for the phosphorylation of aromatase and thus its enzyme activity in in vitro experiments in line 268.

  • Lines 318-320. The final concentration should be mentioned, it does not make sense to indicate volumes. The substrate concentrations indicated for the activity measurements are 2.4 and 1.9 nM. Such concentrations are far away from the saturating ones and also from the Km (at least for the human enzyme). How was this concentration chosen? If the CYP19 isoforms have different Km, then isoforms will have different rates at different substrate concentration. The experiment shown in Figure 2 would make sense if the concentration is saturating for all the compared enzymes.

Since we have not used a standard method for the determination of aromatase activity, we have taken great care to describe the method in as much detail as possible, including both the final concentrations and the resulting volumes. We hope that this will enable colleagues who wish to use this method themselves to set it up easily in their own laboratories. For this reason, we do not wish to shorten the description of the method.

We have determined the substrate concentrations in the preliminary experiments mentioned in lines 381-382.

We have now clearly stated the limitations of the measurements we used in the Discussion section (lines 333-341):

“We did not measure enzyme activity by the commonly used tritium-water assay (via the release of 3H water during the aromatase reaction), but separated the steroids chromatographically after the enzymatic reaction and determined the 3H activity of the individual peaks. This approach is both more sensitive and more specific, since the decrease of the substrate can be detected simultaneously with the increase of the specific metabolites. Our measurements provide initial but clear indications of the different enzyme properties of the two aromatase isoform types. However, for a more precise biochemical characterization, further analyses, such as the determination of Vmax and Km, should be performed in future studies.”

We would like to to make it clear that our study does not claim to be a complete and conclusive description of the enzymatic properties of the new isoform type. However, we show here certainly first but clear results of the different enzymatic properties of the type 2 specific aromatase mutants. Despite the limitations of the methodology with respect to a quantitative characterization of the enzyme activity, the conclusions "The mutant CYP19 isoforms show increased androgen conversion due to the KN transmembrane helix" and "In addition, the TY substitutions in helix C result in a substrate preference for androstenedione" appear to be qualitatively justified on the basis of the data obtained, especially in view of the high reproducibility of the results in three completely independent experiments.

  • Lines 338-339. I do not understand here what the meaning of this sentence is.

As described in the Methods section, the incubated samples were separated by HPLC, the eluate was collected in fractions, and the 3H activity (counts per minute) present in each fraction was measured. Substrate peak (androstenedione or testosterone) and product peak (estrone or estradiol) were identified in the chromatograms based on retention times. After subtraction of the technical background, the 3H activities attributable to the substrate and product peaks were calculated. From the values obtained, the percentage conversion of the amount of substrate added to the reaction preparations was calculated. Compared to the tritium-water assay, which only detects the formation of tritium-labeled water as a result of the release of tritium from the substrate (1β position), the method we used allows not only more sensitive monitoring of substrate conversion, but also more specific monitoring, as more detailed information about the product(s) obtained can be obtained.

Round 2

Reviewer 2 Report

Comments and Suggestions for Authors

The Authors have addressed most of my comments/ questions. I am not sure that my comment about the concentration used for the activity assay was clear but the Authors added a comment on the limitations in the Discussion section. The manuscript can be accepted for publication on IJMS.